# Supplemented Infant Formula and Human Breast Milk Show Similar Patterns in Modulating Infant Microbiota Composition and Function In Vitro

**DOI:** 10.3390/ijms25031806

**Published:** 2024-02-02

**Authors:** Klaudyna Borewicz, Wolfram Manuel Brück

**Affiliations:** 1Mead Johnson B.V., Middenkampweg 2, 6545 CJ Nijmegen, The Netherlands; klaudyna.borewicz@mjngc.com; 2Institute for Life Technologies, University of Applied Sciences Western Switzerland Valais-Wallis, 1950 Sion, Switzerland

**Keywords:** infant health, infant formula, microbiome, high throughput sequencing, INFOGEST, batch culture, SCFA, butyrate, bifidobacteria

## Abstract

The gut microbiota of healthy breastfed infants is often dominated by bifidobacteria. In an effort to mimic the microbiota of breastfed infants, modern formulas are fortified with bioactive and bifidogenic ingredients. These ingredients promote the optimal health and development of infants as well as the development of the infant microbiota. Here, we used INFOGEST and an in vitro batch fermentation model to investigate the gut health-promoting effects of a commercial infant formula supplemented with a blend containing docosahexaenoic acid (DHA) (20 mg/100 kcal), polydextrose and galactooligosaccharides (PDX/GOS) (4 g/L, 1:1 ratio), milk fat globule membrane (MFGM) (5 g/L), lactoferrin (0.6 g/L), and *Bifidobacterium animalis* subsp. *lactis*, BB-12 (BB-12) (10^6^ CFU/g). Using fecal inoculates from three healthy infants, we assessed microbiota changes, the bifidogenic effect, and the short-chain fatty acid (SCFA) production of the supplemented test formula and compared those with data obtained from an unsupplemented base formula and from the breast milk control. Our results show that even after INFOGEST digestion of the formula, the supplemented formula can still maintain its bioactivity and modulate infants’ microbiota composition, promote faster bifidobacterial growth, and stimulate production of SCFAs. Thus, it may be concluded that the test formula containing a bioactive blend promotes infant gut microbiota and SCFA profile to something similar, but not identical to those of breastfed infants.

## 1. Introduction

Extensive evidence has shown that breast milk provides the optimal nutrition to infants. Its nutritional composition and the presence of different non-nutritive bioactive factors in human breast milk assure infant survival and healthy development during early life [1]. As such, The World Health Organization recommends exclusive breast feeding for the first six months of life [2]. However, at times when breastfeeding is not possible, infant formula is recommended to safely replace breastfeeding, either partially or completely. 

Nowadays, infant formulas contain key macronutrients like proteins, carbohydrates, and fats that are necessary to cover the nutritional and energy needs of a growing infant [3]. As our understanding of the composition of human milk increase, it is becoming clear that breast milk not only covers the nutritional needs of an infant but contains a wide array of bioactive factors with medicinal qualities that also have a profound role in infant health [4]. Thus, to better mimic the health-promoting functionality of breast milk, it is increasingly common to supplement infant formulas with various functional ingredients that are also found in breast milk, such as probiotics, DHA, lactoferrin, vitamins and minerals, and others [5,6,7].

Besides nourishing an infant, breastfeeding is also the key factor affecting the development of infants’ gut microbiota. The microbiota composition and function are essential for maintaining immunological, endocrine, and neural homeostasis of the host. During infancy and through life, the microbiota plays critical roles in the defense, metabolism, and trophic functions [8,9,10]. Commensal microbes form a barrier against the proliferation and attachment of pathogenic organisms, help to break down indigestible food components, toxins, and drugs, and they take part in vitamin synthesis and iron absorption. The trophic functions of gut microbiota include the growth and differentiation of the epithelial cells lining the intestinal lumen and the homeostatic maintenance of the immune system, including tolerance to food antigens [8,11].

The microbiota composition of breastfed infants is dominated by genus *Bifidobacterium*. *Bacteroides*, *Streptococcus*, and *Lactobacillus*, and a few other anaerobes and small numbers of facultative anaerobic bacteria have been found in stools of breastfed infants [8]. On the other hand, formula-fed infants have microbiota that are more diverse and with lower abundances of *Bifidobacterium* but increased prevalences of *Bacteroides*, Enterococci, *Enterobacteriaceae, Staphylococcus*, anaerobic *Streptococcus,* and *Clostridium* [8,12,13]. Gut colonization with bifidobacteria is also delayed in formula-fed infants. Changes in microbiota associated with formula feeding, for example an increase in the abundance of *Clostridium* spp., and in particular, *C. difficile*, have been associated with a higher risk to develop atopic symptoms like eczema, recurrent wheeze, allergic sensitization, and atopic dermatitis [14,15]. Thus, in recent years, efforts have been made to “close the gap” between infant formula and breast milk around the modulation of gut microbiota, mainly to increase the bifidogenic effect of the formulas [16,17]. 

In the current study, a commercial infant formula containing a blend of bioactive ingredients was pre-digested to simulate passage through an infant’s stomach and a small intestine and then tested in vitro for its microbiota modulatory effect in comparison to human breast milk and a negative control formula without the bioactive blend. The study was performed with the continuing outlook to improve commercially available infant formulas and to provide an effective substitute for breast milk that simulates some of its benefits.

## 2. Results

### 2.1. High-Throughput Sequencing

A total of 36 samples were included in the analyses. Samples were collected at four different timepoints (T = 0, 12, 24, 48 h) from the fermentations using inoculates from three different infants and from three treatment groups (test formula, base formula, and breast milk). Reads were processed using a standard CosmosID pipeline, and the taxonomic classification of OTUs was performed using DADA2′s naive Bayesian classifier with the Silva version 138 database [18]. The analysis resulted in the detection of five different phyla, namely Firmicutes, Proteobacteria, Actinobacteriota, and two low abundance phyla, namely Fusobacteriota and an unknown Bacteria Kingdom phylum. At the genus level, the most abundant taxa were *Streptococcus*, *Raoultella*, *Enterococcus*, *Bifidobacterium*, *Escherichia-Shigella*, and *Actinomyces* (Appendix A). The microbiota composition during the fermentation of the formulas and breast milk was analyzed to evaluate the prebiotic potential of the formula with the bioactive blend and to correlate it with the productions of SCFAs, lactic acid, and succinic acid over time.

The Bray–Curtis dissimilarity matrix was used to measure differences in microbial composition, and a Principal Coordinate Analysis (PCoA) of the bacterial community dynamics showed a directional shift in community composition in relation to incubation time and the type of the fermentation substrate. At 48 h of incubation, breast milk and test formula samples were clustered together and placed away from the base formula samples (Figure 1). A similar segregation of samples was also observed when using the Jacquard similarity coefficient, which accounts for the presence and the absence of microbial groups. These results indicate that both the fermentation duration and the presence of bioactive blend played important roles in shaping the microbial communities in vitro. 

The microbial alpha diversity was determined based on Shannon’s diversity index (Figure 2). Shannon’s diversity index accounts for both abundance and evenness of the species present. At the beginning of the fermentation, all treatments were inoculated with infant fecal samples that were dominated with genera *Streptococcus*, *Raoultella*, and *Enterococcus*. The diversity changed during fermentation and decreased in all groups between T0 and T12, then stabilized from T12 to T24, and showed strong increases between T24 to T48. This coincided with an increase in the *Streptococcus* abundance at T12 and maintained its levels through T24 in all groups, followed by decreased at T48 in the test formula and breast milk, but only a moderate decrease in the base formula. At the end of fermentation, both the number of observed taxa and the community evenness increased, and in the test formula and breast milk, there was a significant increase in genera *Actinomyces*, *Bifidobacterium*, and *Escherichia-Shigella* and subsequent reductions in *Streptococcus* and *Raoultella* (Appendix A). Consequently, Shannon’s diversity indices at T48 were significantly higher in all groups compared to those at earlier timepoints, and Shannon’s diversity index was significantly higher in both breast milk and test formula treatments compared to those of the base formula (*p* < 0.05), while the difference between breast milk and the test formula groups was not significant (*p* > 0.05; Figure 2).

Differences in the relative abundance of genus-level taxa between each treatment group at each timepoint were calculated using the Kruskal–Wallis test. Differentially abundant taxa were reported in the Appendix A.

Genus-level relative abundance data were subjected to multivariate PCA and RDAs. The PCA confirmed the beta diversity results and showed groupings of samples by treatment and sampling timepoint (Figure 3). All treatments showed a slight separation at the start of the fermentation, indicating that there were small differences in each microbiota already at the start of the experiment. During intermediate sampling points, the separation and the migration of samples were very clear for the breast milk treatment, while less apparent for the test formula and base formula treatments. At the end of the fermentation (T48 h), both the test formula and breast milk samples were clustered very closely together, and away from the base formula groups, which also showed more stable microbiota profiles across the timepoints. 

Partial RDA was used to measure the amount of variation in the microbial composition and SCFA composition as explained by the duration of fermentation (T0, 12, 24, 48 h) or by treatment (breast milk, base formula, and test formula). Sampling time could explain 18% and 27% of the variations in microbiota and SCFA composition, respectively, and their effect was significant (microbiota: FDR*p* = 0.002 and SCFA: FDR*p* = 0.001). To measure the overall treatment effect across all timepoints and to minimize the effect of the sampling timepoint, we ran the RDAs with the fermentation duration set as a covariate. The RDAs showed that treatments accounted for 19.97% of the variation in microbiota composition and that the effect of breast milk and base was significant (FDR*p* = 0.004 and FDR*p* = 0.033). The treatment with test formula resulted in an intermediate community composition that was not significantly different from the other treatments (FDR*p* = 0.058; Figure 4a, b). SCFA production correlated with breast milk (butyrate, acetate, lactate, and succinate) and the test formula (propionate, acetate; Figure 4b).

To further analyze the treatment effect at each timepoint, an RDA was also conducted separately for each subset of samples (Figure 5a–d), revealing that as the fermentation continued, the treatment effect was stronger and explained an increasing amount of variation in the microbial composition (T0 = 59.9%, T12 = 70.47%, T24 = 74.15%, T48 = 85.83%). In addition, as the fermentation progressed, the number of differentially abundant genera between the three treatment groups increased, with the microbiota of the test formula and breast milk treatments converging, while that of the base formula group became more dissimilar, as it was also depicted by the taxa vectors in Figure 5a–d. The full list of all differentially abundant taxa, *p*-values, and pairwise group comparisons can be found in Appendix A.

### 2.2. qPCR Using Bifidobacterium spp. Specific Primers

The absolute abundance of *Bifidobacterium* spp. was estimated using qPCR with the standard curves generated using *Bifidobacterium longum* subsp. *infantis* (DSM 20090). All treatments resulted in an increase in counts of *Bifidobacterium* spp. (Figure 6). There was no significant difference between cell counts in the experimental test formula and breast milk at T12 and T48 h. However, the bifidogenic effect was faster in test formula samples, where the highest concentration of *Bifidobacterium* spp. was obtained already after T24, while in the breast milk and the base formula, the maximum cell count was seen only after T48 h. The mean cell count values at the end of the experiment at T48 were similar for the test formula and breast milk and were 9.23 × log10 CFU/mL for the test formula and 9.14 × log10 CFU/mL for breast milk and were significantly higher than in the base formula (8.78 × log10 CFU/mL; Figure 6).

### 2.3. Short-Chain Fatty Acid Analysis

Immediately after inoculation, succinate and lactate were the predominant SCFAs in the batch culture fermentations, which was a SCFA profile that we expected in feces of healthy, predominately breastfed infants (Figure 7, Table 1(a)). During the fermentation process, the changes in microbiota composition and the increase in *Bifidobacteria* spp. corresponded with changes in the SCFAs. At T24 h, succinate disappeared, and at T48 h, lactate disappeared. The largest amount of acetate was observed in fermentations using digested breast milk as a medium. Breast milk also had the largest accumulation of butyrate. Overall, the average cumulative SCFA concentrations at T48 in the breast milk and test formula treatments were similar and were significantly higher (*p* < 0.05) than in the base formula (Table 1(b)). 

## 3. Discussion

### 3.1. Batch Culture Fermentations

Multiple samples from each donor were collected and frozen, and then all samples were pooled to assure sufficient volume for the in vitro experiment. While fresh feces are often used for in vitro fermentations, freezing fecal samples was previously shown to preserve the viability, short-chain fatty acids concentration, and microbiota composition without significant alteration for up to 12 months [19,20,21]. As fecal samples were stored for a maximum of 3 months in this study, we anticipated no significant changes through the freezing process compared to using fresh samples while allowing us to obtain enough fecal material for the in vitro assays. The use of fecal inoculum batch cultures prepared either from single donors or a pool of donors has been a source of debate among experts [22]. However, the use of pooled fecal inocula reduces inter-donor variabilities, while having a homogeneous and equilibrated mixture reduces the bias usually found with infant feeding regimes and the mode of delivery. In addition, pooled an frozen fecal samples eliminate the problems with sample availability and reduces sample processing labor. Furthermore, being able to use the same inoculum throughout several sets of experiments provides more reliable and reproducible results [23].

### 3.2. Microbiota Variation during Fermentation

Early life colonization patterns of the gastrointestinal tract could significantly contribute to the long-term healthy and unhealthy consequences during an individual’s lifespan [24]. As such, the intestinal microbiota is actively involved in gut maturation and immune system development, particularly in the first 6 months of life [20]. Results from 16S rRNA high-throughput gene sequencing showed that the phyla Firmicutes, Proteobacteria, and Actinobacteria were dominant in the fermentations using pooled and frozen fecal inocula. We observed that the fecal inoculum was dominated by *Streptococcus*, *Raoultella* (*Klebsiella*), and *Enterococcus*, which are commonly found in C-section-delivered babies and are associated with hospital environments [25]. As our pool contained a C-section-delivered infant, this might explain the observed difference. Furthermore, the absence of a fecal inoculum dominated by bifidobacteria may be due to the feces of a 6-month-old infant that was also used in the pool. At the end of the fermentation at T48 h, the test formula and breast milk groups showed significant increases in the genera *Actinomyces*, *Bifidobacterium*, and *Escherichia-Shigella* with subsequent reductions in *Streptococcus* and *Raoultella* (*Klebsiella*). Forty-eight hours is the typical maximum incubation time for batch systems when simulating the large intestine of monogastric animals [26].

Surprisingly, the increase in *Bifidobacterium* spp. as seen via HTS and qPCR was not primarily due to the addition of BB-12 in the test formula, but it was due to an increase in *Bifidobacterium longum* subsp. *infantis* (*B. infantis*) that was present in the fecal inoculum. *B. infantis* is unique in its capacity to consume human milk oligosaccharides (HMOs) as well as two major gangliosides on the surface of fat globules in human milk, GM3 and GD3 [27]. Gangliosides are involved in numerous biological processes such as neuronal development, host–pathogen interactions, and gastrointestinal health and are also present in the milk fat globule membrane (MFGM) of whole bovine milk [28]. The experimental test formula used in this study was fortified with MFGM. In previous studies, the addition of MFGM to infant formula did not increase the similarity between feces from infants fed an MFGM-supplemented formula compared to feces from infants that were breastfed [29]. This was in contrast to our study, although we cannot exclude the possibility that this beneficial effect in our study could be attributed to the presence of other bioactive components in the experimental formula, specifically the prebiotic PDX/GOS. 

The bifidogenic effect of PDX/GOS has been well demonstrated in adults and infants in numerous studies [30,31,32,33]. Scalabrin et al. (2012) specifically showed that if PDX/GOS was added to infant formula, *B. infantis*, *B. longum*, and *B. catenulatum* increased in healthy term infants during a 60-day feeding period [33]. Most importantly, the addition of PDX/GOS to formula increased the amount of total *Bifidobacterium* spp. compared with the control group. Total *Bifidobacterium* spp. did not vary compared to the breastfed group [33]. A similar effect was observed by Fanaro et al. (2009) [31]. The addition of MFGM also showed some evidence of increasing *Bifidobacterium* spp. in infant stool samples [34].

Conversely, a supplementation of *Bifidobacterium animalis* subsp. *Lactis* BB-12, even at high doses, did not affect the numbers of total bifidobacteria in the stool [27]. Hence, the activity of the BB-12 (at 10^6^ CFU/kg) introduced to the infant intestine through the test formula may be limited to previously documented activities such as improving bowel function, and immune health effects such as protection against diarrhea without further growth [35]. 

Similarly, the addition of certain long-chain fatty acids has also been associated with the microbiota development of infants. As such, DHA (C22:6), as it was contained in the test formula, has been positively correlated with increases in *Bacteroides*, *Veillonella*, *Streptococcus*, and *Clostridium* in infants [36,37]. The other genera that were shown to be predominant in all fermentations, namely *Streptococcus* and *Escherichia/Shigella*, in addition to *Veillonella*, were previously identified as the core taxa in infant feces in The INSPIRE Study, being present in 98.4, 91.7, and 90.2% of all samples [38]. These genera were introduced in the infant gut through breast milk, which has the core genera of *Streptococcus* and *Staphylococcus*, although substantial variability exists [38]. Breast milk from mothers that had recent C-section deliveries have a higher relative abundance of Proteobacteria, with a reciprocal reduction in Firmicutes [39]. As the breast milk used in this study came from a mother with a recent C-section, the higher than anticipated abundance of Enterobacteriaceae in our fecal inoculum may be explained.

The microbial alpha diversity based on Shannon’s diversity index showed that the diversity initially decreased in all groups before stabilization and finally showed a strong increase between T24 to T48 h. Additionally, Shannon’s diversity index at T48 was significantly higher in both breast milk and supplemented test formula treatments compared to base formula (*p* < 0.05), while the difference between breast milk and the test formula groups was not significant (*p* > 0.05). In earlier studies, alpha diversity was shown to be lower in breastfed infants compared to formula-fed infants during the first 3 months after birth but increased significantly at 6 months of age. A lower alpha diversity and reduced number of observed species may potentially be beneficial for the infant and may allow the prioritization of *Bacteroides* colonization in later stages of microbiota development [40].

Conversely, it has been speculated that a higher bacterial diversity in formula-fed infants leads to a shift toward an adult-like microbiome at earlier ages, whereas a lower diversity in breastfed infants was mainly due to the abundance of *Bifidobacterium* spp. [41]. As similar temporal trend in alpha diversity was described by Chichlowski et al. (2023) [42]. The authors of the study showed a biphasic change in decreasing alpha diversity from 2 weeks to 3 months of infant age that was followed by an increase in alpha diversity until 6 months of age in exclusively breastfed infants. This biphasic trend was explained by the utilization of bioactive milk components, such as HMOs, at different timepoints and different components of the microbiome. In the present study, the shaping of the microbiome and alpha diversity changes in the batch culture fermentations may also be due to the various bioactive components in the breast milk and the test formula [40].

### 3.3. Short-Chain Fatty Acid Analysis

The fatty acid profile of human milk varies in relation to maternal diet, particularly in long-chain polyunsaturated fatty acids (LCPUFAs). LCPUFA intake in the Western world is skewed toward omega-6 fatty acids, with a sub-optimal intake of omega-3 fatty acids. The docosahexanoic acid (DHA) composition of human milk is particularly low in North American populations; supplementation should be considered for breastfeeding North American women with DHA-limited diets [1].

Short-chain fatty acids (SCFAs) are the main products of anaerobic microbial fermentation in the large intestine and affect colonic health by providing energy to the epithelial cells [43]. In infants, during breastfeeding (early phase), the SCFA profile is characterized by low acetate and high succinate; during complementary feeding (middle phase), by high lactate, pyruvate, and formate; and after cessation of breastfeeding (late phase), by high propionate and butyrate [44]. In this study, early on in the fermentation, the fermentations were dominated by lactate and succinate, which have previously been positively correlated with amounts of fecal water [29]. In addition, lactate has been positively associated with breastfeeding, leading to gut microbiota that is typically dominated by lactic acid-producing bacteria [45]. The presence of lactate in the lumen of infant gastrointestinal tracts may prevent the overgrowth of pH-sensitive pathogenic bacteria, such as *Enterobacteriaceae* and Clostridia [46].

Succinate, a four-carbon dicarboxylic acid (C_4_H_6_O_4_) can be synthesized biologically, both as an intermediate of the TCA cycle and as one of the mixed-acid fermentation products of bacteria during anaerobic metabolism. Various anaerobic and facultative anaerobic bacteria, such as *E. coli*, *Enterococcus flavescens*, *Acinetobacter succinogenes*, *Anaerobiospirillum succiniciproducens*, *Mannheimia succiniciproducens*, and the yeast *S. cerevisiae*, produce succinate as a fermentation product [47]. Previous studies showed that succinate accumulation is associated with an oversupply of complex substrates, such as prebiotics or when the further metabolization of succinate is unnecessary [48]. It is also possible that the lack of Vitamin B12, which is necessary for propionate production, results in the accumulation of succinate instead of propionate [49]. Both lactate and succinate are efficiently utilized by certain groups of anaerobic gut bacteria and are thus considered only intermediate metabolites in the microbial production of acetate, butyrate, and propionate [50,51]. Amongst these SCFAs, acetate often reaches the highest concentration of any of the SCFAs in feces under normal circumstances [45]. Instead of acetate representing a stable endpoint in gut fermentation, it may further be utilized to form butyrate via reductive acetogenesis [52]. Butyrate has significant effects on the development and gene expression of intestinal cells [43]. The highest concentration of butyrate in this study was found in fermentations using breast milk after 48 h, whereas the test formula and the base formulas had butyrate only as a minor component at all timepoints. Acetate was the predominant acid in all fermentations after inoculation, but it was highest in breast milk fermentations, which is in line with previous studies [53]. High levels of acetate were also obtained in fermentations with the test formula. However, levels of acetate concentrations like those found in fermentations with breast milk were not obtained. 

Our study has some limitations, the main one being a small sample size of the replicates. Another limitation is the lack of controls for each individual bioactive that would allow us to separate the individual effects of each and to determine the possible synergic interactions within the blend. Further studies, especially clinical studies including different populations of infants, for example those born via C-section, or infants receiving antibiotic treatments, would allow us to better understand the formula effect under different starting microbiota conditions. Long-term follow-up clinical studies would also provide data on the long-term clinical effects of test formula use in pediatric populations. 

## 4. Materials and Methods

The formulations tested in this study included the following: a supplemented test formula (Mead Johnson B.V., Nijmegen, The Netherlands) following the global standard composition of infant formula [54], containing a blend of bioactive components, and reconstituted according to the manufacturer’s instructions; base formula (Mead Johnson B.V., Nijmegen, The Netherlands)—a negative control formula without the bioactive blend; breast milk—a positive control. The bioactive blend contained DHA (20 mg/100 kcal), PDX/GOS (4 g/L, 1:1 ratio), MFGM (5 g/L), lactoferrin (0.6 g/L), and BB-12 (10^6^ CFU/g). Breast milk was collected in 100 mL bags between 6 and 9 weeks of lactation and frozen until use. All liquid formulations were reconstituted in distilled water, aseptically to minimize risk of contamination.

Infant fecal samples from three different infants were used as microbial sources for the in vitro fermentation experiment. The infants were between 2 and 6 months old and fed predominantly breast milk with supplemental formulae. Neither of the infants had any complications during birth or experienced any significant past diseases or used medication that would affect the study. All infants were full-term (born between 39 weeks, 0 days and 40 weeks, 6 days) with one 2-month-old delivered vaginally, one 2-month-old delivered via C-section, and one 6-month-old delivered vaginally. The parents were asked to collect soiled diapers as soon as possible and place them in an anaerobic jar (Oxoid AnaeroJar, AG0025, Thermo Fisher Diagnostics AG, Basel, Switzerland). A sachet (AnaeroGen AN0025, Thermo Fisher Diagnostics AG, Basel, Switzerland) was placed in each jar and sealed, where the atmospheric oxygen in the jar was rapidly absorbed to below 1%/30 min, with the simultaneous generation of carbon dioxide at an expected level between 9% and 13% and without the need of water activation. The jars were collected within an hour of the parents giving notice to the laboratory staff. The jars were opened in an anaerobic chamber (Whitley A85 Workstation, Don Whitley Scientific Limited, Bingley, UK), and fecal samples were scraped off into a sterile 50 mL falcon tube. The fecal samples were aliquoted and mixed with glycerol (20%) in a 50:50 *w*/*v* proportion and shock-frozen individually in liquid nitrogen before storage at −80 °C until further use [19].

Frozen breast milk samples were obtained from the mother of the 2-month-old delivered via c-section and expressed using a breast pump. The milk sample was expressed directly into breast milk storage bags (Medela Schweiz AG, Baar, Switzerland) at least two hours after the previous feed. Creams, ointments, and soaps on the breasts and nipples prior to collecting the milk samples were avoided when possible. Breasts were also thoroughly washed with soap and rinsed with water prior to the collection. After collection, the breast milk samples were immediately frozen at −20 °C until further use.

### 4.1. INFOGEST Digestion

All materials were standard analytical grade. Chemicals, enzymes, and bile acids were purchased from Sigma Aldrich (Buchs, Switzerland). Pepsin (Sigma P6887) and pancreatin (Sigma, P7545 8XUSP) were of porcine origin, while bile (Sigma B8631) was of bovine origin. Rabbit gastric extract (RGE) was provided by Lipolytech (Marseille, France). The parameters of the digestion were as part of the INFOGEST COST action [55,56]. The infant gastrointestinal in vitro batch model was set up to mimic, as close as possible, the digestive conditions of full-term infants. Aseptic techniques were used throughout all assays.

In the first step of the digestion, all milks (test formula, base formula, and breast milk) were subjected to the gastric phase of the INFOGEST protocol. Gastric phase parameters (meal to secretions ratio, pH) were determined based on the infant gastric conditions occurring at the emptying half-time, assumed to be more representative than the final timepoint. As described by Bourlieu et al. (2014), a gastric emptying half-time of 78 min has been reported for infant formula [55]. A compilation of the data measuring gastric pH in infants allowed the determination of a linear regression describing the gastric acidification curve, pH = −0.015 × time (min) + 6.52, as previously described [55]. Considering the gastric emptying half-time of 78 min, gastric pH in the static model was set up at 5.3. Based on postprandial enzyme activities determined in infant gastric aspirates, average values of 63 U of pepsin and 4.5 U of lipase per mL of gastric content and per kg of body weight of infant [57,58] were found. Thus, considering the mean body weight of a one-month-old infant of 4.25 kg [59], enzyme activities were set up at 268 U/mL of gastric content for pepsin, and 19 U/mL of gastric content for lipase. Pepsin and gastric lipase were added as rabbit gastric extract (RGE). Rabbit gastric lipase presents 85% of the sequence homology compared to the human one [60]. The added amount of RGE covered 100% of the pepsin activity and 110% of the lipase activity required (21 U/mL). The gastric fluid composition was based on a study on 30 full-term infants reported by Hyde (1968) [61]. The simulated gastric fluid (SGF) was composed of sodium chloride and potassium chloride fixed at 94 and 13 mM, respectively, and adjusted to pH 5.3 with HCl 1 M. After 60 min of gastric digestion, the pH was increased to 7 through the addition of NaOH 1 M to stop gastric enzyme activities before further intestinal digestion.

The milks digested in the gastric step were then subjected to the intestinal phase. The intestinal phase used a vial containing the 60 min gastric phase adjusted to the intestinal pH of 6.6 using HCl 1 M. The simulated intestinal fluid (SIF), based on the characterization of duodenal fluid of 1-week-old full-term infants [62], was composed of 164 mM of sodium chloride, 10 mM of potassium chloride, and 85 mM of sodium bicarbonate and adjusted at pH 6.6. Calcium chloride was added separately before the beginning of the intestinal phase at a concentration of 3 mM within the volume of the intestinal fluid [62]. Bovine bile extract was added to a final content of 3.1 mM of bile salts, which corresponds to the average postprandial value determined in duodenal aspirates of eight 2-week-old infants [63]. The added amount of pancreatin covered the intestinal lipase activity required of 90 U/mL of intestinal content [64] and that the trypsin activity needed i.e., 16 U/mL of intestinal content, which was consistent with previously reviewed in vivo data [65].

### 4.2. Batch Culture

After the INFOGEST digestions, the in vitro fermentations were carried out in anaerobic batch culture systems. All samples were adjusted to pH 6.5 and a 4 mL/L of a 250 mg/L stock solution of resazurin (Fisher Scientific, Reinach, Switzerland), and 0.5 g/l L-cysteine-HCl (Sigma, Prod. Code: C1276-10G) was added. Due to the probiotic culture content of the test formula, all digested milks were not sterilized, and aseptic techniques through all assays were used to minimize the contamination risk. In total, 90 mL of the prepared milks were added to 100 mL batch culture vessels and maintained under an atmosphere of oxygen-free nitrogen gas by continuously sparging with oxygen-free nitrogen (15 mL/min). The vessels were magnetically stirred, and pH was maintained using a pH controller (Electrolab Biotech Limited, Gloucestershire, UK) [66]. Fecal slurries were prepared anaerobically at 10% (*w*/*v*) using anaerobic phosphate-buffered saline (0.1 M, pH 6.5) and homogenized for 2 min at 460 paddle beats (Stomacher 400, Seward, West Sussex, UK). The maximum time for sample preparation after the reception of the fecal sample and inoculation of fermenter vessels was 15 min. The final dilution factor of the fecal sample was 1:100. Experiments were performed in triplicate. Batch cultures ran under anaerobic conditions for a period of 48 h; during which, samples (5 mL) were collected at T = 0 h, 12 h, 24 h, and 48 h for high-throughput sequencing, and T = 0 h, 6 h, 12 h, 24 h, and 48 h for (short-chain fatty acid) SCFA analyses. The T0 samples were taken from their respective vessels under operating conditions and not from volunteer fecal slurries. Samples were stored at − 80 °C until further analysis. 

### 4.3. High-Throughput Sequencing

DNA extraction and 16SrRNA sequencing were carried out on 250 mg of aliquoted samples using the QIAGEN DNeasy PowerSoil Pro Kit (47014, Qiagen, Hilden, Germany), according to the manufacturer’s protocol. DNA concentrations were quantified using a Qubit 4 fluorometer and Qubit™ dsDNA HS Assay Kit (Thermo Fisher Scientific, Basel, Switzerland). For the PCR, DNA extracts were normalized to 5 ng/µL with PCR-certified water. The 25-μL PCR mix was composed of 5 ng/μL DNA, 1 × FastStart PCR grade nucleotide mix buffer without MgCl_2_, 4.5 nM MgCl_2_, 200 μM of each PCR grade nucleotide, 0.05 U/μL Fast Start Taq DNA Polymerase, 400 nM target-specific primers, 5% DMSO, and 9 μL of PCR-certified water. The PCR cycling conditions consisted of an initial activation step at 95 °C for 3 min, followed by 32 cycles with denaturation at 95 °C for 30 s, annealing at 62 °C for 30 s, extension at 72 °C for 30 s, and final extension at 72 °C for 10 min. The V3-V4 region of the 16 S ribosomal RNA (rRNA) gene was amplified in a total volume of 1 µL containing 5 ng of template using uniquely barcoded primers: 341F-n (5′-CCTACGGGNGGCWGCAG-3′) and 805R-n (5′-GACTACHVGGGTATCTAATCC-3′) [67]. Sequencing was performed on an Illumina MiSeq platform 2 × 250 bp.

A bioinformatics analysis was carried out using the CosmosID-HUB interface. The CosmosID-HUB Microbiome’s 16S workflow implements the DADA2 algorithm [68] as its core engine and utilizes the Nextflow ampliseq pipeline [69] definitions to run it on cloud infrastructure. Briefly, primer removal was performed with Cutadapt [70], and quality trimming parameters were passed to DADA2 to ensure the median quality score over the length of the read exceeded a certain Phred score threshold. Within DADA2, forward and reverse reads were each trimmed to a uniform length based on the quality of reads in the sample. DADA2 uses machine learning with a parametric error model to learn the error rates for the forward and reverse reads, based on the premise that correct sequences should be more common than any error variant. DADA2 then applies its core sample inference algorithm to the filtered and trimmed data, applying these learned error models. Resulting paired-end reads were then merged if they had at least 12 bases of overlap and were identical across the entire overlap. The resulting table of sequences and observed frequencies was then filtered to remove chimeric sequences (those that exactly match a combination of more prevalent “parent” sequences). Taxonomy and species-level identifications (where possible) were conducted with DADA2′s naive Bayesian classifier, using the Silva version 138 database [18]. Gut microbiota diversity was assessed using alpha and beta diversities, with Shannon’s diversity index used to summarize the microbial community, where a higher alpha diversity indicates a greater number of species, with more even representation. The statistical difference between treatments at each timepoint was calculated using the Wilcoxon test in the CosmosID-HUB interface. The beta diversity was calculated using Bray–Curtis dissimilarity matrix to examine the similarity of communities between samples. 

Unconstrained (PCA) and constrained (RDA) multivariate analyses were carried out on log-transformed genus-level relative abundance data in Canoco5 [71]. In the resulting plots each point corresponded to one sample, and the proximity of samples indicates a higher similarity in taxa relative abundance. The taxa vectors point toward the samples in which the relative abundance of these taxa was the highest, while the vector lengths correspond to the R-squared measure calculated by dividing the taxa scores by their SD [71]. For better visibility, only the ten best-fitting taxa were plotted and any differentially abundant taxa that were identified with Kruskal–Wallis analyses were also included on the plots. The best-fitting taxa were those that explained the highest percentages of variation in the relative abundance within the ordination axes. In the RDA analyses, the explanatory variables included the sampling timepoint or the experimental treatment group. The significance of the explanatory variables was assessed using the Monte Carlo permutation test at 499 random permutations [71]. Differentially abundant genus-level taxa were identified using the Kruskal–Wallis analysis in QIIME [72].

### 4.4. qPCR Using Bifidobacterium spp. Specific Primers

A 1 mL aliquot of each of the batch culture samples was extracted using the QIAGEN DNeasy PowerSoil Pro Kit (Qiagen), according to the manufacturer’s protocol. DNA samples were quantified using a Qubit 4 fluorometer and Qubit™ dsDNA HS Assay Kit (Thermo Fisher Scientific). The amplification of *Bifidobacterium* spp. was performed according to a protocol adapted from Rinttilä et al. (2004) [73]. A 243 bp product was amplified using primers F: 5′-TCGCGTC(C/T)GGTGTGAAAG-3′ and R: 5′-CCACATCCAGC(A/G)TCCAC-3′. The final reaction mixture contained a 1:75,000 dilution of SYBR Green I (Thermo Fisher Scientific), 10 mm Tris-HCl (pH 8.8), 150 mm KCl, 0.1% Triton X-100, 3 mm MgCl_2_, 100 μm of each dNTP, 0.5 μm of each primer, 0.6 U Dynazyme II polymerase (Finnzymes #F-501L, Thermo Fisher Scientific, Basel, Switzerland)), and either 5 μL of template or water. The qPCR conditions were as follows: 95 °C for 5 min followed by 35 cycles of 95 °C for 15 s, 58 °C for 20 s, 72 °C for 30 s, and 80–85 °C for 30 s to collect the fluorescent data. A melting curve analysis was performed by slow cooling from 95 to 60 °C, with fluorescence collection at 0.3 °C intervals and a hold of 10 s at each decrement. Each qPCR run of samples was run concurrently with DNA extracts containing between 7.6 × 10^9^ CFU/mL and 4.6 × 10^6^ CFU/mL of *Bifidobacterium longum* subsp. *infantis* (DSM 20090) to generate a standard curve. Culture cell densities were determined from cultures grown in Bifidobacteria Selective Broth (BSM-B, Merck 90273-500G-F, Darmstadt, Germany) anaerobically for 48 h at 37 °C using a Neubauer-improved counting chamber with a chamber depth of 0.02 mm (Assistent, Glaswarenfabrik Karl Hecht GmbH & Co KG, Sondheim vor der Rhön, Germany). Each standard and sample were run in triplicate. 

### 4.5. Short-Chain Fatty Acid Analysis

Short-chain fatty acid (SCFA) analyses were performed on supernatants obtained after the 1 mL fermentation aliquots had been centrifuged at 13,000× *g* for 10 min and then syringe-filtered through a sterile 0.45 µm syringe filter. The filtrates were then injected directly into an HPLC (Agilent, Basel, Switzerland) using a Cation H+ pre-column and Rezex ROA-Organic Acid H+ (8%) column (30 × 4.6 mm, Phenomenex, Aschaffenburg, Germany), and run with an isocratic mobile phase of 5 mM H_2_SO_4_ at a flowrate of 0.5 mL min^−1^ (62 min, 70 °C). Lactic acid, acetic acid, succinic acid, propionic acid, and butyric acid were detected via refractive index and UV (210 nm) spectrometry. The Shapiro–Wilk Test online calculator (https://www.statskingdom.com/shapiro-wilk-test-calculator.html, accessed on 29 November 2023) was used to test the resulting data for normality, and the Kruskal–Wallis test was used to measure the statistical differences for each individual acid and the cumulative SCFA concentration. To aid visualization, replicate samples were averaged and displayed together [66,74]. 

## 5. Conclusions

Based on the results presented here, it may be concluded that the test formula containing a bioactive blend of DHA (20 mg/100 kcal), PDX/GOS (4 g/L, 1:1 ratio), MFGM (5 g/L), lactoferrin (0.6 g/L), and BB-12 (10^6^ CFU/g) promotes an infant’s gut microbiota and SCFA profile to become similar, but not identical to those of breastfed infants. However, only feeding studies in infants may give conclusive evidence to the formula’s effectiveness in improving the progression of the infant gut microbiota and other health outcomes. 

## Figures and Tables

**Figure 1 ijms-25-01806-f001:**
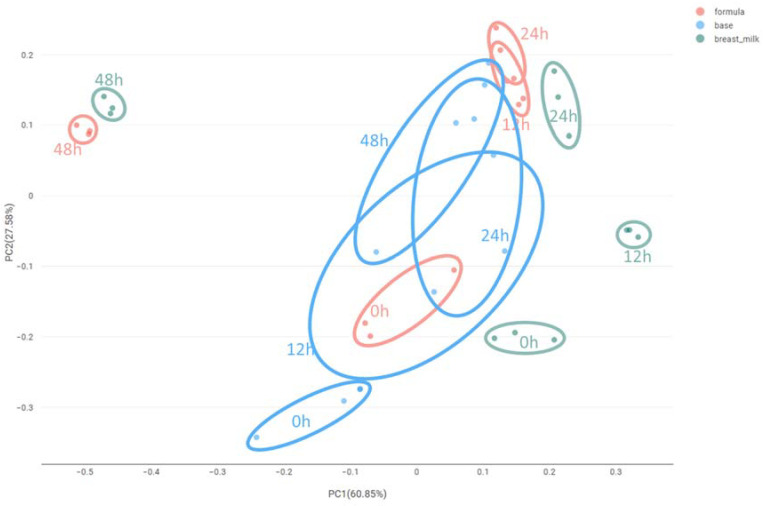
PCoA based on the Bray–Curtis dissimilarity matrix showing differences between observed microbial communities at T0, 12, 24, and T48 h during the in vitro fermentation using infant fecal inocula as microbial sources and the supplemented formula, base formula, and breast milk as substrates.

**Figure 2 ijms-25-01806-f002:**
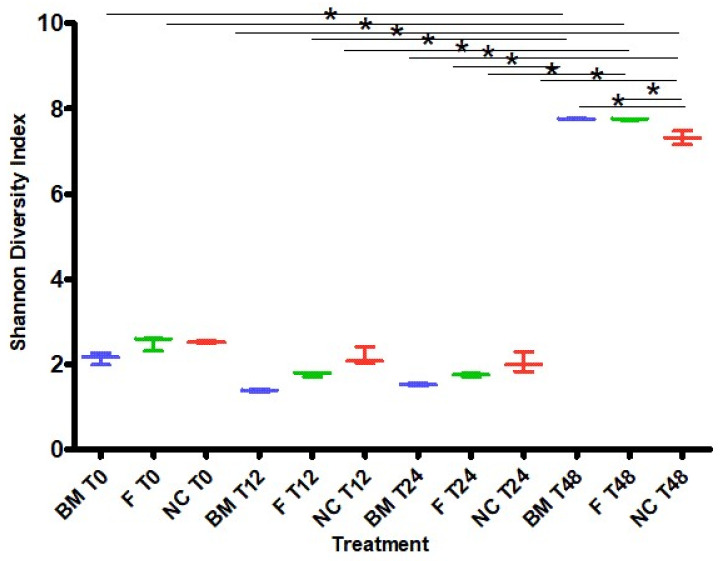
Shannon diversity index values describing the microbial alpha diversity after a T0 h, 12 h, 24 h, and 48 h fermentation. Samples were fermented using infant fecal inoculates, and test formula (F), breast milk (BM), and base formula (NC) were used as substrates. Significant differences were identified between the test formula and base formula at T48 (*p* < 0.05). The difference between breast milk and the test formula groups at T48 was not significant (*p* > 0.05). * indicates statistically significant difference (*p* < 0.05).

**Figure 3 ijms-25-01806-f003:**
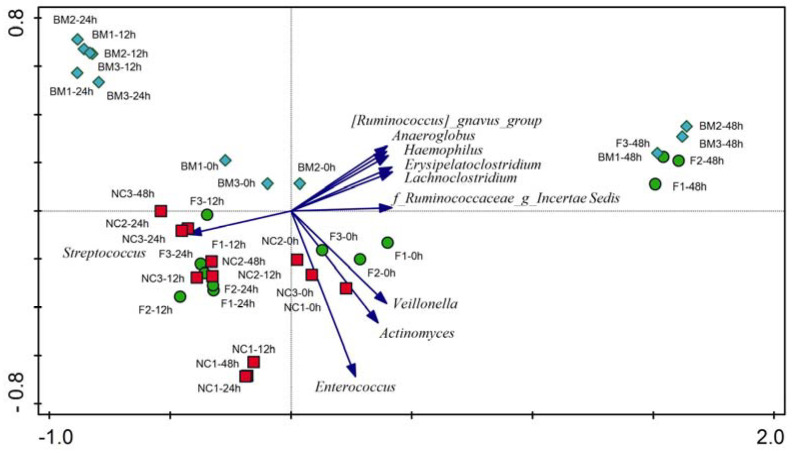
PCA on log-transformed genus-level relative abundance data. Samples are colored by treatment group (green—test formula; red—base formula; blue—breast milk). For clarity, the ten best-fitting genera and their corresponding vectors are displayed. The direction of vectors points towards higher relative abundance of a given genus.

**Figure 4 ijms-25-01806-f004:**
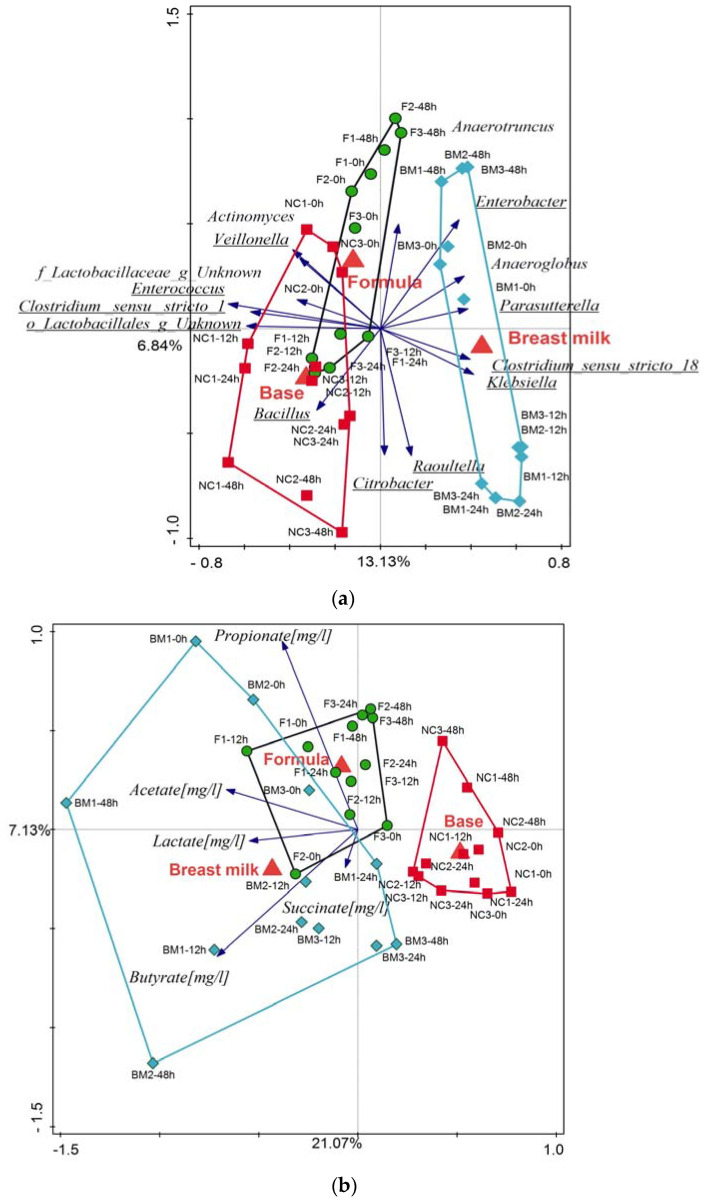
Partial RDAs with fermentation times as a covariate and using (**a**) log-transformed genus-level relative abundance and (**b**) SCFA concentrations. Samples are color-coded by treatment group (green—test formula; red—base formula; blue—breast milk). For clarity, the ten best-fitting genera and their corresponding vectors are displayed (**a**)**.** Any additional genera that were differentially abundant in the Kruskal–Wallis analysis are also included in the figure and indicated by underline (**a**). The direction of blue vectors points towards higher relative abundance of the genus (**a**) or a higher concentration of each SCFA (**b**).

**Figure 5 ijms-25-01806-f005:**
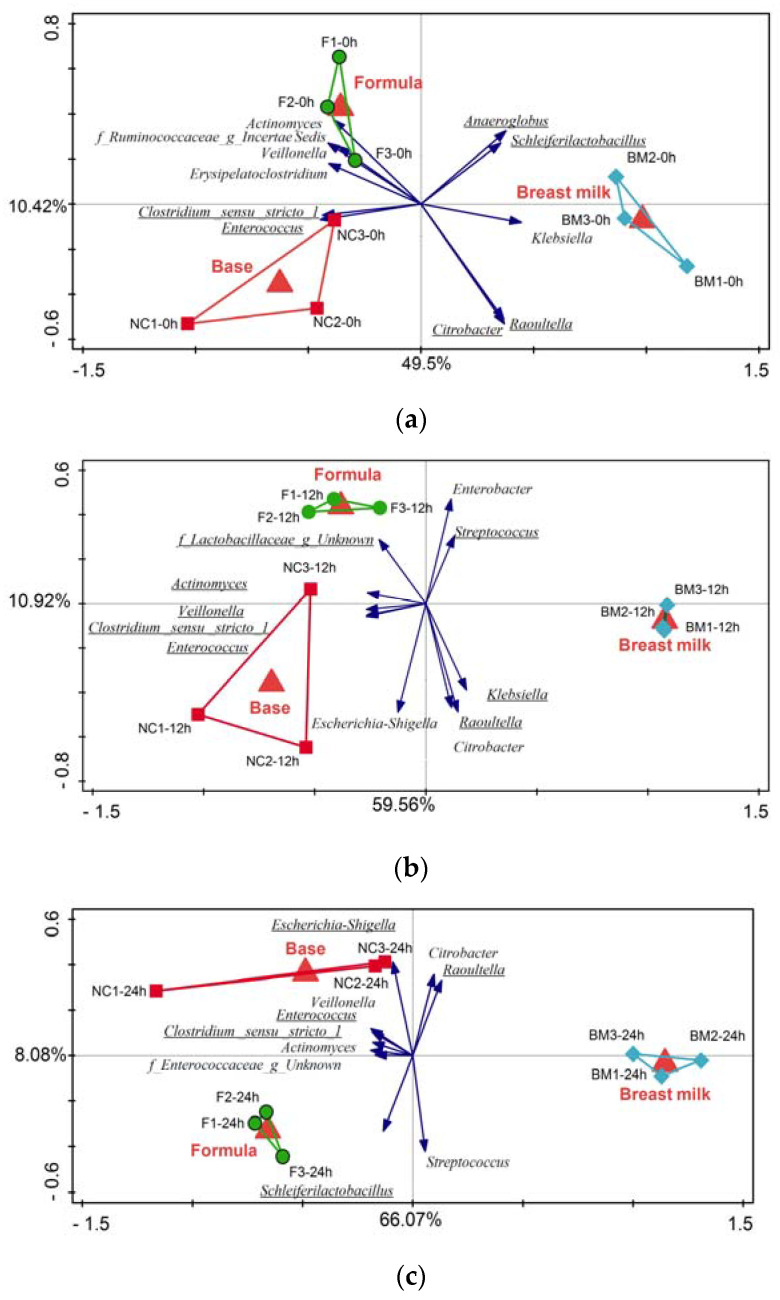
RDAs using the log-transformed genus-level relative abundance data at each timepoint; (**a**) T0 h; (**b**) T12 h; (**c**) T24 h; (**d**) T48 h. Samples are colored by treatment group (green—test formula; red—base formula; blue—breast milk). For better clarity, only the ten best-fitting genera and their corresponding vectors are displayed. The direction of blue vectors points towards higher relative abundance of a given genus. Additional genera that were differentially abundant in the Kruskal–Wallis analysis are included in the figure and are indicated by the underline.

**Figure 6 ijms-25-01806-f006:**
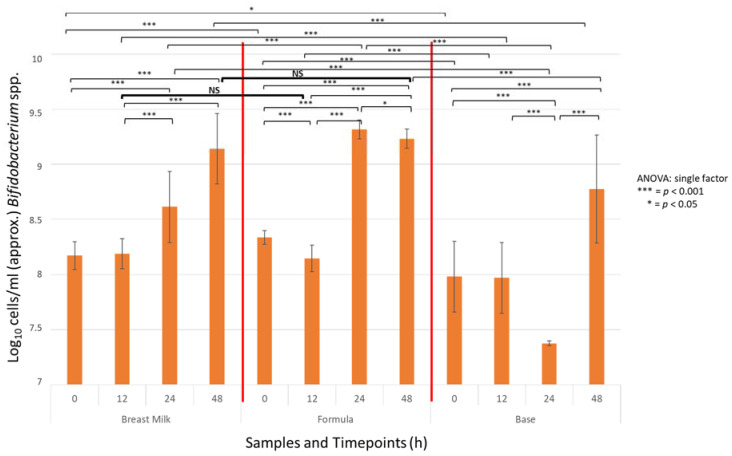
qPCR counts of *Bifidobacterium spp.* (±SD) in batch cultures containing either the test formula, breast milk, or base formula inoculated with a mixture of infant feces at T0 h, 12 h, 24 h, and 48 h after inoculation. Statistically significant differences between groups are shown. *** = *p* < 0.001, * = *p* < 0.05. NS - Non-Significant difference (*p* > 0.05).

**Figure 7 ijms-25-01806-f007:**
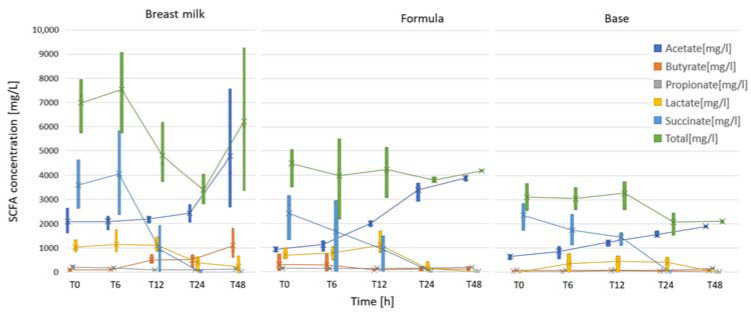
Concentrations of volatile fatty acids in each treatment at each sampling timepoint.

**Table 1 ijms-25-01806-t001:** Average concentrations of individual and total volatile fatty acids in each treatment at each sampling timepoint. The intermediate sampling point at T = 6 h is also included. (a) Group mean concentrations in mg/L; (b) Kruskal–Wallis results indicating similarity between treatments including pairwise comparisons where treatment names are abbreviated as follows: F—test formula treatment, BM—breast milk, B—base formula. NS—Non-Significant difference (*p* > 0.05). nan—no value possible.

**(a)**																		
**Avg.** **Concentrations**	**T0**			**T6**			**T12**			**T24**			**T48**			**All Timepoints**		
SCFA	Formula	Base	Breast milk	Formula	Base	Breast milk	Formula	Base	Breast milk	Formula	Base	Breast milk	Formula	Base	Breast milk	Formula	Base	BM
Acetate [mg/L]	936	655	2088	1125	862	2090	2000	1243	2188	3396	1553	2437	3900	1898	4784	2271	1242	2717
Butyrate [mg/L]	296	31	83	283	13	93	68	64	495	121	44	501	107	56	1093	175	42	453
Lactate [mg/L]	693	0	1030	780	360	1143	1103	440	1097	137	403	380	0	0	220	543	241	774
Propionate [mg/L]	142	74	196	130	74	156	138	85	81	151	70	85	189	153	124	150	91	128
Succinate [mg/L]	2423	2353	3597	1667	1743	4063	937	1443	940	0	0	0	0	0	0	1005	1108	1720
Total [mg/L]	4491	3114	6995	3985	3053	7545	4245	3275	4800	3804	2070	3403	4196	2108	6221	4144	2724	5793
**(b)**																		
** *p* ** **-values**	**T0**			**T6**			**T12**			**T24**			**T48**			**All Timepoints**		
SCFA	All groups	FvBM	FvB	All groups	FvBM	FvB	All groups	FvBM	FvB	All groups	FvBM	FvB	All groups	FvBM	FvB	Time Effect	Treatment Effect	
Acetate [mg/L]	0.027	0.050	0.050	NS	0.,050	NS	NS	NS	0.050	0.03	0.050	0.050	NS	NS	0.050	0.000	0.000	
Butyrate [mg/L]	NS	NS	0.050	NS	NS	NS	NS	0.050	NS	NS	NS	NS	NS	0.050	NS	NS	0.000	
Lactate [mg/L]	0.046	NS	0.037	NS	NS	NS	NS	NS	0.050	NS	NS	NS	NS	NS	nan	0.002	0.008	
Propionate [mg/L]	0.039	NS	0.050	NS	NS	0.050	NS	0.050	0.050	0.04	0.050	0.050	NS	0.050	NS	NS	0.000	
Succinate [mg/L]	NS	NS	NS	NS	NS	NS	NS	NS	NS	nan	nan	nan	nan	nan	nan	0.000	NS	
Total [mg/L]	0.039	0.050	NS	NS	0.050	NS	NS	NS	NS	NS	NS	0.050	NS	NS	0.050	NS	0.000	

## Data Availability

Metagenome raw sequencing datasets are available at the National Center for Biotechnology Information under the BioProject ID: PRJNA1071977.

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
