# Peer review of "Supplemented Infant Formula and Human Breast Milk Show Similar Patterns in Modulating Infant Microbiota Composition and Function In Vitro"

_ijms, 2024, doi:10.3390/ijms25031806_

Round 1
Reviewer 1 Report
Comments and Suggestions for Authors
In this study, Borewicz et al. found that the supplemented infant formula and human breast milk modulated infant microbiota composition and function in vitro in similar patterns. After carefully reviewing this paper, I recognized there were a few issues. Here are some comments on this paper:
1. There are some formatting issues in the manuscript, such as line 17 “Milk” should be “milk”, line 20 “SCFA” should be “SCFAs”, line 60 “Enterococci” needs to be italicized, line 193 “9.23*log10”, Table 1 should be a three-line table, line 372 “100ml” should be “100 mL”, line 441 “Ninety ml”, and line “0,02mm”.
2. It is proposed that the authors add a concluding sentence at the end of the abstract.
3. The significance and importance of this study should be added in the introduction section.
4. It is better to provide a flow schematic to make it easier for the readers to understand how the experiment was performed.
5. In Figures 3 and 7, it is recommended that significant differences be labeled with the letters (a,b,c…).
6. Can the authors explain the reasons why the Shanonn index increased significantly at 48 h?
7. In the Author Contributions section, it is recommended to state who is responsible for the conduct of the experiment.
Author Response
Dear Editor,
We appreciate the editor’s and anonymous reviewer’s comments and believe to have answered the comments sufficiently below for the paper to be reconsidered for publication in the International Journal of Molecular Sciences. All changes in the documents have been left as track changes.
Reviewer 1:
In this study, Borewicz et al. found that the supplemented infant formula and human breast milk modulated infant microbiota composition and function in vitro in similar patterns. After carefully reviewing this paper, I recognized there were a few issues. Here are some comments on this paper:
- There are some formatting issues in the manuscript, such as line 17 “Milk” should be “milk”, line 20 “SCFA” should be “SCFAs”, line 60 “Enterococci” needs to be italicized, line 193 “9.23*log10”, Table 1 should be a three-line table, line 372 “100ml” should be “100 mL”, line 441 “Ninety ml”, and line “0,02mm”.This has been changed. However, as “Enterococci” is not an official Genus (Enterococcus), we have left the term non-italicized according to official nomenclature.
- It is proposed that the authors add a concluding sentence at the end of the abstract. This has been added.
- The significance and importance of this study should be added in the introduction section. This has been added.
- It is better to provide a flow schematic to make it easier for the readers to understand how the experiment was performed. We created a flow chart as graphical abstract which is now included in the paper.
- In Figures 3 and 7, it is recommended that significant differences be labeled with the letters (a,b,c…). The authors thank the reviewer for all their valuable comments. However, we feel that the separation of significance within and between the groups based on the *, **, *** system is also visually appealing and gives a quick overview to the level of significance.
- Can the authors explain the reasons why the Shanonn index increased significantly at 48 h? The Shannon index measures the number of species according to their relative evenness. We observed that as between 24 and 48 h sampling, the population became more even, which could explain the changes in the Shannon index. This was particularly the case in the supplemented formula and breast milk fermentations. In addition to the evenness and number of observed taxa as can be seen in the supplementary table (relative abundance data). We speculate that once the prebiotic compounds are depleted, the taxa that grew fast at first lose their competitive advantage, allowing other taxa can grow. This may be one of the reasons that most publications presenting batch culture data stop their fermentations after 24h.
- In the Author Contributions section, it is recommended to state who is responsible for the conduct of the experiment. An investigation part has been added to the contributions section. In addition, an acknowledgement section has been added for the technical personnel that carried out the study.
Reviewer 2 Report
Comments and Suggestions for Authors
The aim of the study was to determine the effect of commercial infant formula containing a mix of different bioactive compounds on infant microbiota composition and SCFA production, using INFOGEST and in vitro batch fermentation.
The study is interesting, but some improvements are needed before publication.
The terms “in vitro” and “in vivo” should appear in italics throughout the manuscript
Material and methods:
Were breast milk samples obtained from the mothers of the infant donors? Please give more detail about the mothers and milk sampling. Was the breast milk microbiota living at the time of batch fermentation?
The rationale for using infant donors of different postnatal age (2 and 6 months) and type of delivery (vaginal and C-section) must be explained. Why not use more homogenous samples?
A table with the nutritional composition of both formulae should be added as a supplementary file.
Results
Are there any results on the Infogest step?
In the material and Methods section, it was described that bifidobacterial cell densities were determined by culture on selective medium. However, the corresponding results were not shown. It would be interesting to know what is the survival of BB12 in the batch fermentation step.
What was the proportion of BB12 in the increase of bifidobacteria observed with the test formula?
In legend of Figure 1, specify the color corresponding to each group
Figure 3 apparently corresponds to Figure 2 (and successively to the following figures). The abbreviation NC which appears in the X-axis of this figure is not explained in the legend.
Are there differences by phylum?
In the Discussion section, it must be detailed whether DHA, MFGM, and lactoferrin can affect the gut microbiota.
Author Response
Dear Editor,
We appreciate the editor’s and anonymous reviewer’s comments and believe to have answered the comments sufficiently below for the paper to be reconsidered for publication in the International Journal of Molecular Sciences. All changes in the documents have been left as track changes.
Reviewer 2:
The aim of the study was to determine the effect of commercial infant formula containing a mix of different bioactive compounds on infant microbiota composition and SCFA production, using INFOGEST and in vitro batch fermentation.
The study is interesting, but some improvements are needed before publication.
- The terms “in vitro” and “in vivo” should appear in italics throughout the manuscript. This has been changed.
Material and methods:
- Were breast milk samples obtained from the mothers of the infant donors? Please give more detail about the mothers and milk sampling. Was the breast milk microbiota living at the time of batch fermentation? Details on the breast milk donor and the sampling procedure were added to the materials and methods section. We did not check if the breast milk microbiota was living at the time of batch fermentation. However, the microbiota of the breast milk was also analysed using 16S rDNA sequencing.
- The rationale for using infant donors of different postnatal age (2 and 6 months) and type of delivery (vaginal and C-section) must be explained. Why not use more homogenous samples? The authors wanted to avoid a bias related to age and delivery as the infant formula is intended for all infant in the age groups sampled. Using a homogenous sample would thus not be representative of the target groups.
- A table with the nutritional composition of both formulae should be added as a supplementary file. Unfortunately, the composition cannot be disclosed at this time as the formula is currently undergoing patenting.
Results
- Are there any results on the Infogest step? We have not looked at the outflow of the Infogest step as that was not the goal of this study.
- In the material and Methods section, it was described that bifidobacterial cell densities were determined by culture on selective medium. However, the corresponding results were not shown. It would be interesting to know what is the survival of BB12 in the batch fermentation step. We only estimated cell densities of the reference culture for qPCR by incubation on selective medium followed by cell counting in a hemacytometer to prepare standard concentrations for the qPCR. We did not estimate bifidobacterial counts from the batch culture by culture. However, we estimate that the large majority of the bifidobacterial population in the formula was eliminated by the Infogest step. Hence, in our discussion we suggest that:
“Surprisingly, the increase of Bifidobacterium spp. as seen by HTS and qPCR was not primarily due to the addition of BB-12 in the test formula, but it was due to an increase of Bifidobacterium longum subsp. infantis (B. infantis) which was present in the fecal inoculum.”
- What was the proportion of BB12 in the increase of bifidobacteria observed with the test formula? Please refer to our answer above.
- In legend of Figure 1, specify the color corresponding to each group. This has been done.
- Figure 3 apparently corresponds to Figure 2 (and successively to the following figures). The abbreviation NC which appears in the X-axis of this figure is not explained in the legend. This has been done.
- Are there differences by phylum? The differences between the phyla composition between the three groups were similar to the differences observed in the genus comparison. Hence we have decided to present the genus information in this paper.
- In the Discussion section, it must be detailed whether DHA, MFGM, and lactoferrin can affect the gut microbiota. The effect of DHA, MFGM on the infant microbiota have now been added to the discussion. Recent evidence (DOI: https://doi.org/10.1128/spectrum.00096-23) suggest that oral bovine lactoferrin does not impact the diversity or composition of the infant gut microbiota.
Round 2
Reviewer 1 Report
Comments and Suggestions for Authors
Figure 7 would be more concise if the significant differences were denoted with letters.
I have no further concerns.